# Water Profitability Analysis to Improve Food Security and Climate Resilience: A Case Study in the Egyptian Nile Delta

**Adham Badawy** [1,2,*], **Amgad Elmahdi** [1], **Sayed Abd El-Hafez** [3] **and Ali Ibrahim** [3]

1. MENA Office, International Water Management Institute (IWMI), Giza 12661, Egypt; amgad.elmahdi@gmail.com
2. Department of Earth & Environment, Boston University (BU), Boston, MA 02215, USA
3. Soil, Water and Environmental Research Institute (SWERI), Giza 12613, Egypt; s.a.abdelhafez@gmail.com (S.A.E.-H.); dr.ali_66@yahoo.com (A.I.)
* Correspondence: adhamb@bu.edu

**Abstract:** The food self-sufficiency policy has always featured as an unquestionable policy objective for Egypt. This is understandable when one considers both the high population growth and the social and political vulnerability associated with a dependence on food imports and world market food prices such as wheat. Intensive agriculture has led to a growing subsidy burden for the Egyptian government. In addition, the agricultural fields in Egypt are commonly distributed with relatively small sizes parcels that usually reduce the reliability of the agricultural sector, particularly in the delta region, to meet the national food policy. On top of that, climate change, through changing weather patterns and increased temperatures, is affecting agricultural yields and thus farmers' livelihoods. A water profitability analysis was conducted for three governorates in the Nile Delta in Egypt to establish a baseline and assess the net return per unit of water of the main crops in each of these governorates; this can act as a reference of the water profitability of different crops before they are affected by climate change and other internal and external factors. The analysis was based on extensive in-person surveys in each governorate in addition to workshop discussions with farmers. The study has highlighted the impact of a lack of extension services, which limits farmers' ability to increase their land and water productivity. Farmers with more access to subsidized production inputs managed to achieve higher levels of water profitability even on smaller lands. Finally, we drew from our findings key policy actions to improve water profitability and land productivity for farmers in the Nile Delta to achieve higher levels of food security. This will help build resilient food production systems that are reliable in the face of climate change and other drivers. In addition, an integrated nexus strategy and plan for the inter- and intra-country is recommended to address the challenges related to food and climate security.

**Keywords:** water profitability; water productivity; Nile Delta; food security; water security

## 1. Introduction

Since the 1950s, the food self-sufficiency policy has always featured as an unquestionable policy objective for Egypt. Climate change can have a severe impact on the agricultural sector and the stability of food security in Egypt. This is understandable when one considers both the high population growth and the social and political vulnerability associated with a dependence on food imports and world market food prices such as that of wheat. Egypt is considered one of the largest importers of wheat and a country where people rely on wheat products for around one-third of their food consumption in terms of calorie intake [1]. It is also expected that the food and water gaps that Egypt is facing will significantly widen by 2050 [2].

Food security, job creation, and limited per-capita land endowment in the Old Lands were always the determining factors for water and agricultural policy and are constantly

used as an indisputable rationale for the expansion of irrigation, as illustrated in the Ministry of Agricultural and Land Reclamation's (MALR) sustainable development strategy towards 2030 [3]. Moreover, the responsibility of MALR is to ensure that food production is sufficient for to meet demand and sustainable at the same time, in addition to the monitoring and evaluation of sudden climatic changes and their impact on crop productivity to mitigate climate impacts on the quality and productivity of crops under stress.

Climate change can have a severe impact on the agricultural sector and the stability of food security in Egypt and in the Middle East and North Africa (MENA) region [4]. It is expected that crop production will be affected negatively due to the expected increases in temperature, extreme weather events, drought, plant diseases, and pests. Additionally, land use will be affected due to seawater intrusion and salinization. Water resources will be affected due to global warming and decreases in precipitation. Moreover, crop water requirements are expected to increase [5]. The compound effect of all these components represents the main challenge for researchers; moreover, the current cropping systems must be changed to comply with the future demands of the growing population and the threat of climate change [6]. The negative impacts of climate change on crop production can be reduced by the implementation of integrated farm-level adaptation strategies, starting with adopting changes including different seed varieties, planting dates, rationalizing the use of water and fertilizers, and changing irrigation intervals.

In addition, the sustainable development goals (SDGs) 1, 2, and 6, which are promoting sustainable agricultural practices to end poverty and increase water use efficiency [7], need greater efforts and resources at the country level to ensure the even and equal achievement of targets [8]. Therefore, further efforts are required to face these challenges, including more investments in agricultural and food systems and adapting sustainable alternative crops to the impact of water scarcity and climate change. For the sake of rationalizing the use of resources in the agricultural production system in Egypt, there is a need to understand the agricultural system (crops) and its related costs, returns, and profitability for farmers in terms of both land and water.

Water profitability analysis for policy planning—while still relatively a new concept—has been conducted in multiple regions to assess the net return per unit of water consumed in agriculture for crop production. In the Middle East and North Africa (MENA) region, several of these studies were performed. Oulmane et al. [9] assessed the water productivity and water value of three crops under normal conditions and used water-saving technologies in Algeria. The water value was calculated using gross margin, water costs, and applied water. It highlighted the increase in net returns per cubic meter of water due to the use of water-saving technologies. In Jordan, ref. [10] conducted a multicriteria analysis for water productivity to evaluate the economic value of water under maximum yields for selected crops. The study showed date palm to be the most profitable crop regarding water productivity. In Lebanon, a water profitability analysis was conducted to optimize cropping patterns based on the net revenue per unit of water [11]. In Oman, Al-Said et al. [12] assessed the water productivity of vegetables under modern irrigation methods. They analyzed the income per unit of water for five different crops and showed the increased returns and savings gained using drip irrigation for vegetable production.

Further, economic water productivity has been assessed by scholars following the water footprint concept [13]. Chouchane et al. [14] analyzed the economic water and land productivity of 11 crops in Tunisia. The study highlighted that the highest economic water productivity was reported for tomatoes and potatoes, while the lowest was recorded for olives, which are one of the major export products of the nation. In Pakistan, a study was conducted to compare the water productivity and return per unit of water for different rice types [15].

Yakubu et al. [16] analyzed net farm income per unit of land for four major strategic crops in the Kano River irrigation project in Nigeria. The study highlighted the profitability of maize, rice, and wheat compared to tomatoes. Tashikalma et al. [17] compared the crop profitability per unit of land under both rainfed and irrigated conditions in Nigeria. The

study highlighted the major inputs that are costly for farmers, which, if subsidized, could drastically improve their incomes. Khansa [18] analyzed the average farm income under normal cropping patterns and used alternative saving crops. The author showcased the potential increase in farm income and water savings by changing the cropping pattern. Similar studies were conducted in Turkey [19], Peru [20], Bolivia [21], and Mexico [22].

However, the study of water profitability has been scarcely implemented in Egypt and thus there are few data on the baseline for its assessment in the Nile Delta. The only study analyzing water profitability in Egypt was conducted by Hosni et al. [23], who assessed the economic value of water used in irrigation in three governorates. Furthermore, they used linear programming (LP) to optimize the cropping patterns of these governorates to maximize water profitability and water savings. However, Osama et al. [24] studied the net return obtained per unit area (feddan) of all allocated crops for the cropping pattern (2008–2012). They used a linear programming (LP) technique to optimize the area allocated for each crop to achieve an overall increase in net benefits. These studies were conducted using reported data and lacked local farmers' information and voices.

This study attempts to set a baseline for the water profitability of multiple strategic crops in the Egyptian Nile Delta and compares different crops in three different governorates based on primary data collected from farmers' input. These crops were selected due to their importance in terms of the cultivated area, food insecurity, economy, and employment in Egypt. The analysis sheds the light on the main factors contributing to the heterogeneity of the water profitability levels across the Nile Delta and what policy recommendations and actions could be followed to increase and improve water and land profitability and productivity for farmers.

## 2. Materials and Methods

### 2.1. Description of the Study Area

This study was conducted in three governorates in the Egyptian Nile Delta, in Sharkia, El-Beheira, and Kafr El-Sheikh governorates, as seen in Figure 1. The three governorates were selected as they are a good representative of the 'old lands' in the Egyptian Delta, where most smallholder farmers are situated. The three governorates also represent the east, west, and middle of the Delta. The study area is representative as many of the strategic crops in Egypt are grown in these governorates.

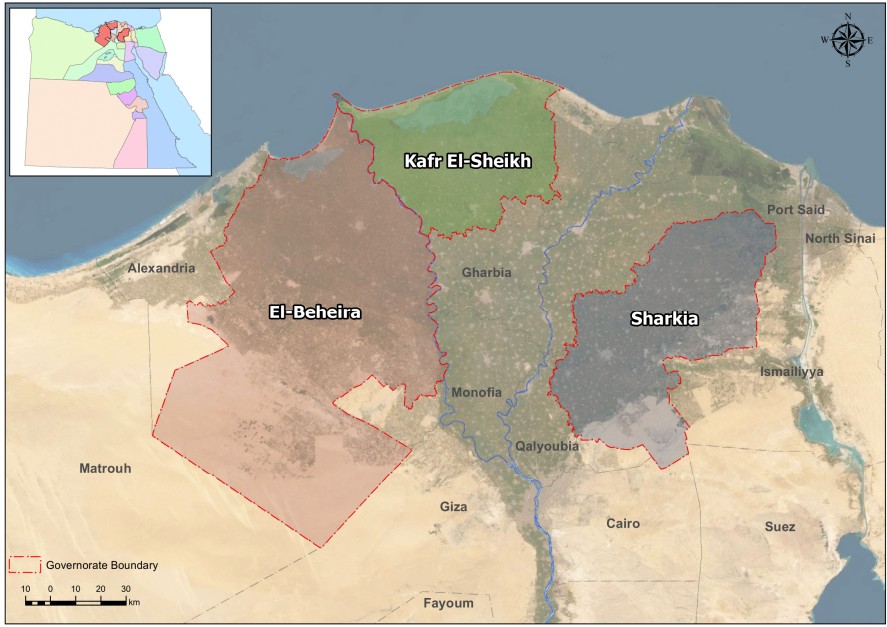

**Figure 1.** Study area in the Egyptian Nile Delta: Sharkia, El-Beheira, and Kafr El-Sheikh governorates. Source: Google Earth.

## 2.2. Data Collection

The study operationalizes both quantitative and qualitative data. The data were collected through two approaches. Firstly, structured interviews for farmers through a structured questionnaire were conducted. For each governorate, a minimum of 100 farmers were interviewed during 2020 to establish the baseline before COVID-19. Secondly, farmer consultation workshops were conducted in each governorate to collect the information and data required but also to validate the data collected during the individual interviews. Three workshops were conducted, one at every governorate with a minimum of 20 farmers. Farmers were randomly selected from each governorate to represent smallholder farmers that have farms ranging from less than one feddan up to five feddans. Table 1 shows the number of farmers in each data collection approach.

**Table 1.** Number of forms collected and farmers attending the workshops.

| Governorate | Number of Forms Collected | Number of Participants in the Workshop |
| --- | --- | --- |
| Sharkia | 102 | 22 |
| El-Beheira | 110 | 20 |
| Kafr El-Sheikh | 120 | 20 |

The questionnaire used for data collection included both qualitative and quantitative data [25]. It included socio-economic data, farmer family structure, a detailed breakdown of all production inputs and their cost, water consumption, agricultural yields and productivity, self-consumption, market access, and selling prices.

The production input costs included:

- Land preparation;
- Seeding and planting;
- Irrigation;
- Fertilization;
- Weeding;
- Pest Control;
- Harvesting;
- Transportation;
- Other Expenses.

## 2.3. Analytical Methodology

Assessing water profitability in this study was achieved using the economic water productivity analytical method [12,23]. This analytical method was chosen due to several reasons. Firstly, the data collected and questionnaires were designed to follow the same structure of production input classification as the agricultural statistics bulletin. Moreover, the data collected allowed for collecting actual water applied by farmers. Finally, the chosen analytical method can accommodate the nuances and differences between farmers regarding access to agricultural inputs and water application compared to statistical averages and experts' estimations. The following steps describe the calculation method:

- **Total Costs (TC):** This is the summation of all the production costs.

$$\text{Total Costs} = \sum(\text{Land Preparation} + \text{Seeding and Planting} + \ldots + \text{Other Expenses.}) \quad (1)$$

- **Total Revenue (TR):** the yield per unit area multiplied by the selling price of all crops on land including primary and secondary crops.

$$\text{Total Revenue} = \text{Yield} \times \text{Crop Price} \quad (2)$$

- **Net Return (NR):** This is the difference between the total revenue and total costs.

$$\text{Net Return} = \text{TR} - \text{TC} \tag{3}$$

- **Water Applied (WA):** the amount of water applied per unit area for that crop production per season.
- **Water Profitability (WP):** the net return per unit of water applied for that crop's production.

$$\text{Water Profitability} = \text{NR/WA}$$

Units for results:

- EGP (Egyptian pound);
- Feddan, unit of area commonly used in Egypt, equivalent to 4200 m$^2$.

## 3. Results

This section presents the results of the collected data and the water profitability analysis conducted for the crops identified by sampled farmers as crops they had planted As the smallholder farmers were randomly selected, data for some crops are not available for any of the three governorates.

### 3.1. Sharkia

Table 2 depicts the water profitability analysis conducted for Sharkia from the collected forms in the governorate. The analysis was conducted for five main crops: wheat, sugar beet, clover, rice, and maize. Regarding total costs per feddan, sugar beet was the highest followed by rice, and the lowest was clover. Total revenue was highest for sugar beet and clover and lowest for maize. For the net return per feddan, clover was the highest 17,480 EGP/feddan, and the lowest value was found for maize at 723 L.E/feddan. Furthermore, rice is considered to be the most water-intensive crop, requiring about 6480 m$^3$/feddan, which is nearly double the amount required for the other crops, and the least water-intensive is wheat, using 2160 m$^3$/feddan. Finally, regarding water profitability, clover was the most profitable at 5.2 EGP/m$^3$ followed by sugar beet and wheat, while maize was the least water-profitable crop at 0.22 EGP/m$^3$.

**Table 2.** Water profitability analysis for Sharkia governorate in 2020, source: field data collected from farmers and verified by workshops.

| Production Inputs | Wheat | Sugar Beet | Clover | Rice | Maize |
|---|---|---|---|---|---|
| Land Preparation (EGP) | 783 | 1320 | 736 | 1097 | 959 |
| Seeding and Planting (EGP) | 899 | 1575 | 466 | 1634 | 1174 |
| Irrigation (EGP) | 421 | 545 | 589 | 1138 | 608 |
| Fertilization (EGP) | 949 | 2310 | 938 | 1101 | 1472 |
| Weeding (EGP) | 200 | 1700 | - | 521 | 715 |
| Pest Control (EGP) | 261 | 900 | - | 360 | 399 |
| Harvesting (EGP) | 2224 | 2750 | 920 | 2537 | 1414 |
| Transportation (EGP) | 587 | 700 | 854 | 634 | 680 |
| Other Expenses (EGP) | 231 | 200 | 217 | 237 | 239 |
| Total Cost Without Rent (EGP) | 6555 | 12,000 | 4720 | 9257 | 7661 |
| Productivity (Ton/feddan) | 2.82 | 45 | 37 | 3.675 | 2.62 |
| Price (EGP/Ton) | 4400 | 500 | 600 | 3500 | 3200 |
| Revenue (EGP/feddan) | 12,408 | 22,500 | 22,200 | 12,863 | 8384 |
| Net Return (EGP/feddan) | 5853 | 10,500 | 17,480 | 3605 | 723 |
| Water Applied (m$^3$/feddan) | 2160 | 3520 | 3360 | 6480 | 3240 |
| **Water Profitability (EGP/m$^3$)** | **2.71** | **2.98** | **5.2** | **0.56** | **0.22** |

Figure 2 shows the water profitability for the selected crops in Sharkia Governorate calculated using the Ministry of Agriculture and Land Reclamation–Economic Affairs Sector's Agricultural Statistics Bulletin for 2019 and data from the Land and Water Research Institute of the Water Standards Department. The figure also shows that clover is the most water-profitable crop, followed by sugar beet and wheat. The least water-profitable crops are rice and maize, which aligns with the data collected from the surveys and workshops. However, the collected data reflect the real value on the ground that show the sugar beet and wheat values were almost double the published value. These on-ground data need to be reflected in the policy planning and recommendations.

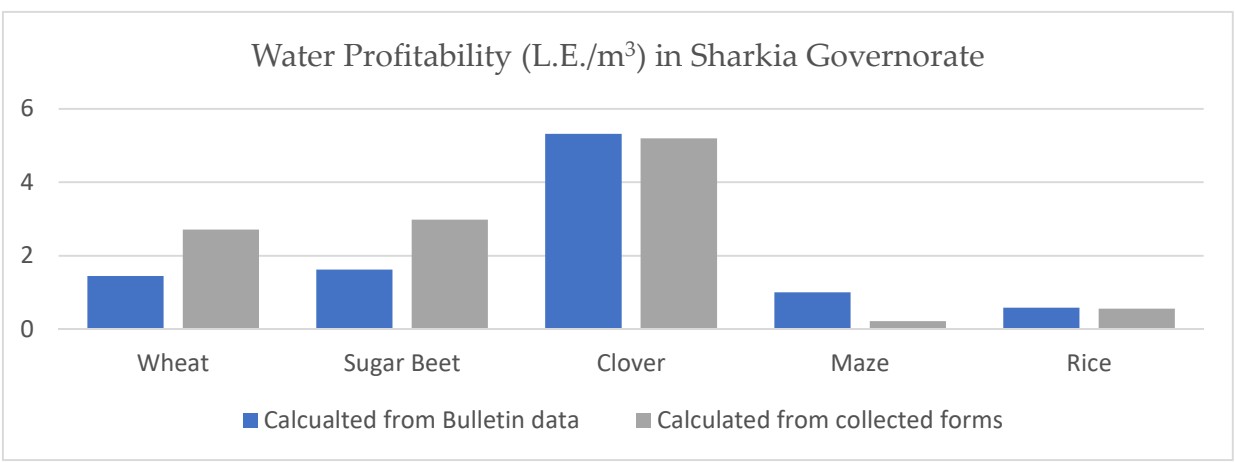

**Figure 2.** Water profitability of Sharkia governorate from Bulletin data compared to collected forms. Source: In blue, Ministry of Agriculture and Land Reclamation–Economic Affairs Sector's Agricultural Statistics Bulletin for 2019 and Land and Water Research Institute's Water Standards Department's unpublished data. In gray are the in-person forms collected in 2020.

*3.2. El-Beheira*

Table 3 depicts the water profitability analysis conducted for El-Beheira from the collected forms in the governorate. The analysis was conducted for nine main crops: wheat, sugar beet, broad bean, clover, rice, maize, watermelon pulp, tomato, and cotton. Regarding total costs per feddan, tomato was the highest followed by cotton, and the lowest was broad bean. Total revenue was highest for cotton and tomato and lowest for maize. For the net return per feddan, watermelon pulp was the highest 19,391 EGP/feddan and the lowest was maize at 10,799 L.E/feddan. Furthermore, rice was found to be the most water-intensive crop, followed by maize. Finally, for water profitability, watermelon pulp was the most profitable at 13.47 EGP/m$^3$ followed by broad bean at 12.96 EGP/m$^3$, while maize was the least water-profitable crop at 1.71 EGP/m$^3$.

Figure 3 shows the water profitability for the selected crops in the El-Beheira governorate calculated using the Ministry of Agriculture and Land Reclamation–Economic Affairs Sector's Agricultural Statistics Bulletin for 2019 and data from the Land and Water Research Institute's Water Standards Department compared to results calculated from the collected forms/study. Moreover, the water profitability values gathered from the collected forms are significantly higher than those from the Bulletin and Water Standards Department. This is reflected in ground farmers' information and local conditions. This analysis shows the potential of cotton and wheat in El-Beheira.

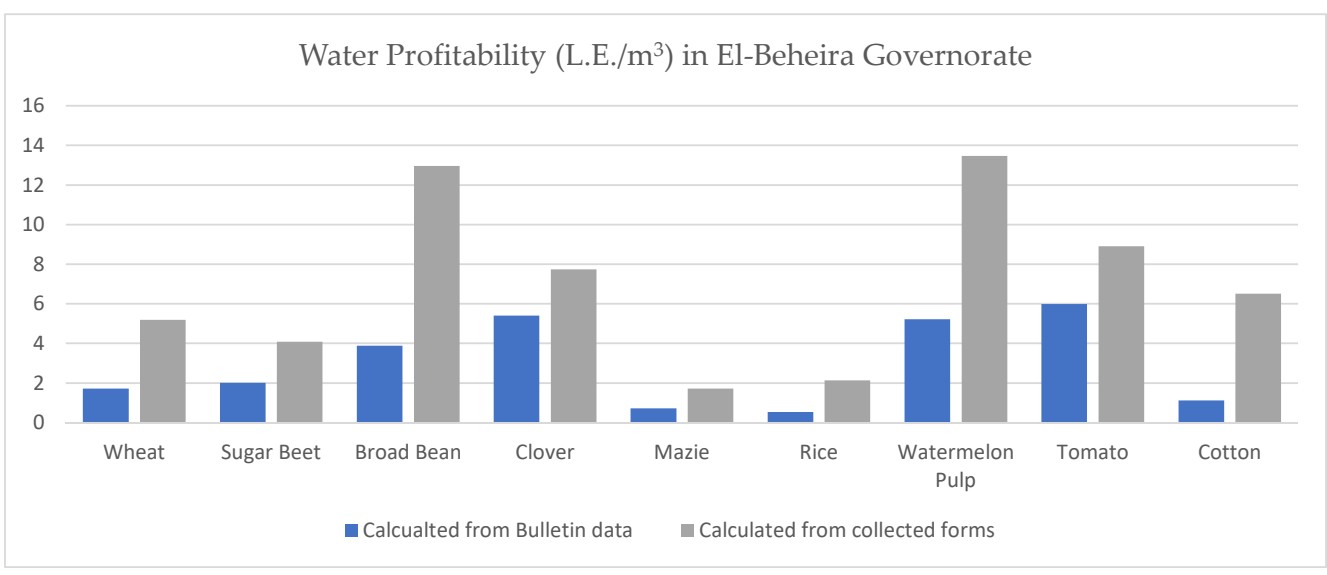

**Figure 3.** Water profitability for El-Beheira Governorate from Bulletin data compared to collected forms. Source: In blue, Ministry of Agriculture and Land Reclamation–Economic Affairs Sector's Agricultural Statistics Bulletin for 2019 and Land and Water Research Institute–Water Standards Department's unpublished data. In gray are the in-person forms collected in 2020.

**Table 3.** Water Profitability Analysis for El-Beheira Governorate in 2020, source: field data collected from farmers and verified by workshops.

| Production Inputs | Wheat | Sugar Beet | Broad Bean | Clover | Rice | Maize | Watermelon Pulp | Tomato | Cotton |
|---|---|---|---|---|---|---|---|---|---|
| Land Preparation (EGP) | 416 | 368 | 294 | 314 | 483 | 548 | 413 | 1280 | 690 |
| Seeding and Planting (EGP) | 656 | 341 | 744 | 597 | 973 | 782 | 370 | 1380 | 730 |
| Irrigation (EGP) | 396 | 600 | 391 | 494 | 682 | 564 | 536 | 360 | 340 |
| Fertilization (EGP) | 858 | 968 | 596 | 624 | 870 | 1018 | 1023 | 2270 | 1060 |
| Weeding (EGP) | 414 | 573 | 175 | 607 | 605 | 506 | 526 | 1160 | 1000 |
| Pest Control (EGP) | 350 | 386 | 225 | 864 | 534 | 474 | 735 | 4700 | 1540 |
| Harvesting (EGP) | 1001 | 1000 | 763 | 1076 | 879 | 838 | 657 | 1340 | 2140 |
| Transportation (EGP) | 269 | 473 | 475 | 166 | 292 | 303 | 130 | 500 | 250 |
| Other Expenses (EGP) | 23 | 67 | - | 21 | 30 | 56 | 21 | 200 | - |
| Total Cost Without Rent (EGP) | 4371 | 4764 | 3661 | 4758 | 5333 | 5060 | 4409 | 13,190 | 7750 |
| Productivity (Ton/feddan) | 3 | 22 | 1.86 | 35 | 4 | 3.36 | 0.7 | 26 | 1.575 |
| Price (EGP/Ton) | 4467 | 650 | 12,000 | 680 | 3700 | 3214 | 34,000 | 1000 | 16,825 |
| Revenue (EGP/feddan) | 13,401 | 14,300 | 22,320 | 23,800 | 14,800 | 10,799 | 23,800 | 26,000 | 26,499 |
| Net Return (EGP/feddan) | 9030 | 9536 | 18,659 | 19,042 | 9467 | 5739 | 19,391 | 12,810 | 18,749 |
| Water Applied (m³/feddan) | 1740 | 2340 | 1440 | 2460 | 4440 | 3360 | 1440 | 1440 | 2880 |
| **Water Profitability (EGP/m³)** | **5.19** | **4.08** | **12.96** | **7.74** | **2.13** | **1.71** | **13.47** | **8.9** | **6.51** |

### 3.3. Kafr El-Sheikh

Table 4 depicts the water profitability analysis conducted for Kafr El-Sheikh from the collected forms in the governorate. The analysis was conducted for ten crops produced in the area: wheat, sugar beet, broad bean, maize, watermelon pulp, clover, cotton, dry peas, and onion. Regarding total costs per feddan, cotton was the highest followed by sugar beet, and the lowest was clover. Total revenue was highest for dry peas followed by onion and cotton and lowest for maize and then clover. For the net return per feddan, dry peas were the highest at 25,000 EGP/feddan followed by onion at 22,544 EGP/feddan, and the lowest was maize at 1697 L.E/feddan. Furthermore, rice was found to be the most water-intensive crop, and the least intensive was dry peas using 1400 m³/feddan. Finally,

for water profitability, dry peas were the most water profitable at 17.86 EGP/m$^3$ followed by onion and broad bean, while maize was the least water-profitable crop at 0.49 EGP/m$^3$.

**Table 4.** Water profitability analysis for Kafr El-Sheikh Governorate in 2020, source: field data collected from farmers and verified by workshops.

| Production Inputs | Wheat | Sugar Beet | Broad Bean | Rice | Maize | Watermelon Pulp | Clover | Cotton | Dry Peas | Onion |
|---|---|---|---|---|---|---|---|---|---|---|
| Land Preparation (EGP) | 616 | 726 | 400 | 691 | 638 | 648 | 389 | 795 | 700 | 700 |
| Seeding and Planting (EGP) | 828 | 823 | 1029 | 1178 | 901 | 841 | 417 | 985 | 375 | 1000 |
| Irrigation (EGP) | 494 | 588 | 236 | 972 | 508 | 371 | 439 | 720 | 425 | 800 |
| Fertilization (EGP) | 942 | 1538 | 414 | 1122 | 1051 | 1064 | 526 | 1520 | 800 | 1500 |
| Weeding (EGP) | 355 | 859 | 643 | 545 | 596 | 580 | 136 | 999 | 625 | 300 |
| Pest Control (EGP) | 424 | 825 | 664 | 747 | 360 | 964 | 231 | 1703 | 850 | 800 |
| Harvesting (EGP) | 850 | 1054 | 1007 | 809 | 644 | 757 | 217 | 2663 | 1000 | 800 |
| Transportation (EGP) | 261 | 573 | 314 | 270 | 306 | 173 | 103 | 248 | 225 | 200 |
| Other Expenses (EGP) | 204 | 300 | 175 | 231 | 200 | 200 | 0 | 250 | 0 | 0 |
| Total Cost Without Rent (EGP) | 4974 | 7286 | 4882 | 6566 | 5204 | 5596 | 2459 | 9884 | 5000 | 6100 |
| Productivity (Ton/feddan) | 2.85 | 27 | 1.395 | 3.25 | 2.1 | 0.7 | 25.5 | 1.339 | 2 | 14 |
| Price (EGP/Ton) | 4467 | 625 | 12,187 | 3560 | 3286 | 35,000 | 400 | 19,810 | 15,000 | 2046 |
| Revenue (EGP/feddan) | 12,731 | 16,875 | 17,001 | 11,570 | 6901 | 24,500 | 10,200 | 26,526 | 30,000 | 28,644 |
| Net Return (EGP/feddan) | 7757 | 9589 | 12,119 | 5004 | 1697 | 18,904 | 7741 | 16,641 | 25,000 | 22,544 |
| Water Applied (m$^3$/feddan) | 2088 | 1740 | 1218 | 5568 | 3480 | 2320 | 2262 | 3712 | 1400 | 1740 |
| **Water Profitability (EGP/m$^3$)** | **3.72** | **5.51** | **9.95** | **0.9** | **0.49** | **8.15** | **3.42** | **4.48** | **17.86** | **12.96** |

　　Figure 4 shows the water profitability for the selected crops in Kafr El-Sheikh calculated using the Ministry of Agriculture and Land Reclamation–Economic Affairs Sector's Agricultural Statistics Bulletin for 2019 and data from the Land and Water Research Institute–Water Standards Department. The figure also shows that dry peas were the most water-profitable crop, followed by onion, while the least-water profitable crops were rice and maize, which aligns with the data collected from the surveys and workshops. Again, the results indicate the potential consideration of cotton and sugar beet in Kafr El-Sheikh.

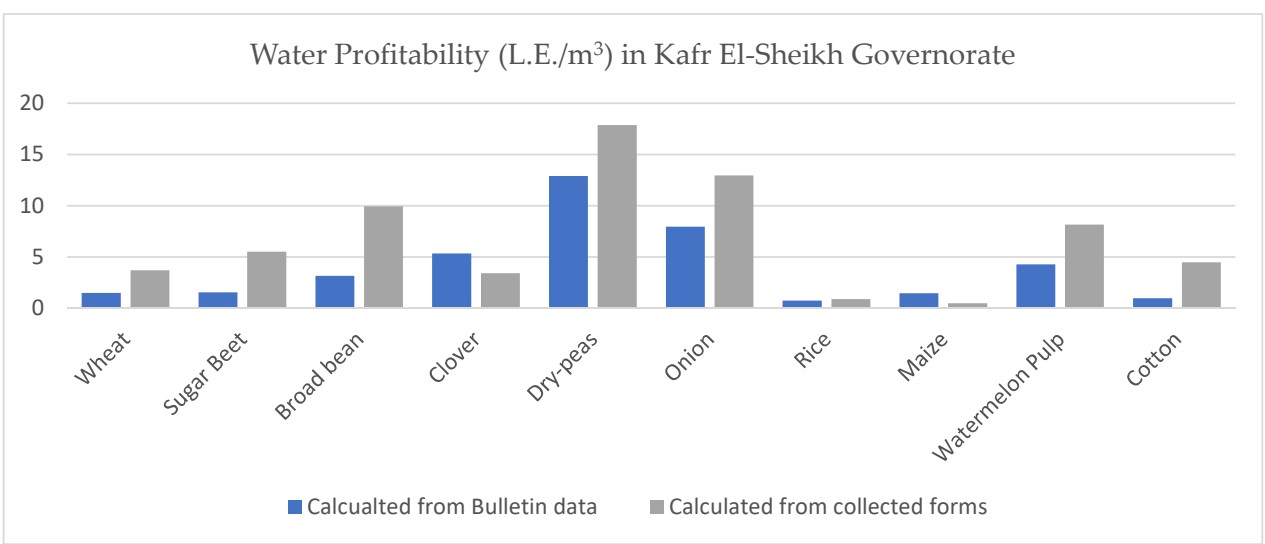

**Figure 4.** Water Profitability for Kafr El-Sheikh Governorate from Bulletin data compared to collected forms. Source: In blue, Ministry of Agriculture and Land Reclamation–Economic Affairs Sector's Agricultural Statistics Bulletin for 2019 and Land and Water Research Institute–Water Standards Department's unpublished data. In gray are the in-person forms collected in 2020.

*3.4. Cross-Governorate Comparison*

From a socio-economic perspective, the results show that the majority of farmers are over 50 years old, as their percentage reached 61% in Sharkia Governorate, about 67% in El-Beheira Governorate, and about 73% in Kafr El-Sheikh Governorate. Additionally, most farmers had received a formal education, as the percentage of educated people reached about 89.22% in Sharkia Governorate, about 66.96% in the El-Beheira Governorate, and about 51% in Kafr El-Sheikh Governorate. The study also showed that agricultural incomes varied from one governorate to another, as the annual income from agriculture reached about 28,000 EGP per feddan in Sharkia Governorate, about EGP 13,000 per feddan in El-Beheira Governorate and about EGP 12,000 per feddan in Kafr El-Sheikh Governorate, and the majority of farmers had incomes other than agriculture.

It was found from the farmers' responses that the most important reason for the low productivity per feddan was the lack of fresh water in the water channels, which forces some farmers to supplement their irrigation needs with water from agricultural drainages such as in Kafr El-Sheikh or well water. Furthermore, across the three governorates, the lack of production requirements such as fertilizers, seeds, pesticides, and machinery needed for cultivation and harvesting operations at appropriate prices and times are increasing the costs and limiting productivity. Furthermore, low market prices are affecting the net returns per feddan and unit of water. In addition, low soil fertility and deteriorating water quality are all contributing to a reduction in productivity and yields.

Figure 5 depicts the water profitability of different crops in each of the three governorates. The highest overall profitable crop was dry peas, which were only found in Kafr El-Sheikh at nearly 18 EGP/m$^3$, followed by watermelon pulp, broad bean, and onion. The least water-profitable crops were rice and maize. However, in El-Beheira wheat was more water profitable than in Sharkia and Kafr El-Sheikh. Furthermore, clover's water profitability value in Sharkia was nearly double that of Kafr El-Sheikh. Moreover, sugar beet in Kafr El-Sheikh was more water profitable compared to the other two governorates. Finally, water profitability was mostly higher for the same crops in Beheira and then Kafr El-Sheikh and lastly Sharkia. These results could be used to reflect on the suitability of the crops per governorate including land and water as well as the socio-economic data of the farmers and accessibility to the market.

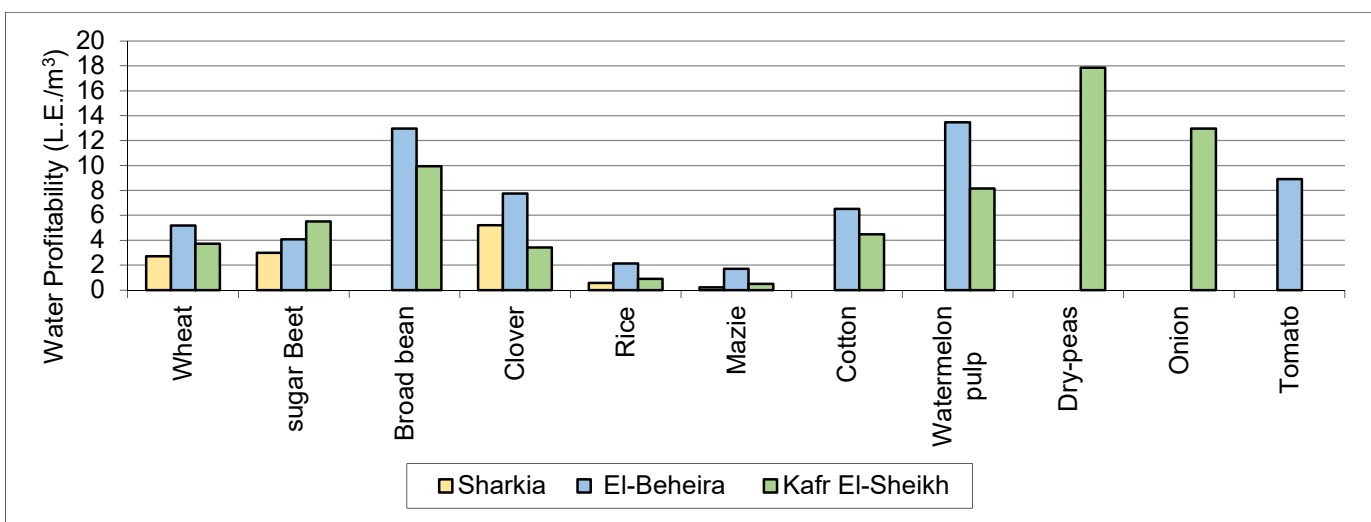

**Figure 5.** Cross-governorate water profitability comparison for 2020, source: field data collected from farmers and verified by workshops.

The collected data also point to the fact that experience had a great impact on increasing water and land productivity and determining the dates of harvest and the best harvesting technique. The level of education also affected production levels and the change of crops from one year to another. The high costs of production requirements had a significant impact on the profitability potential as many farmers did not have the cash flow required to cover the initial costs of the production of the more profitable crop.

The results show that the economic variables vary from one governorate to another, although the data of Sharkia Governorate show higher values than those of other governorates. Thus, the total costs for the same crop are higher in Sharkia compared to El-Beheira and Kafr El-Sheikh. This is one of the main reasons why the net return per feddan and water profitability are lower in Sharkia compared to the other two governorates. Moreover, in Sharkia Governorate the farm sizes are relatively smaller and more fragmented than the other two governorates. This makes the costs relatively higher and the net return per feddan significantly lower. Finally, for smaller farm sizes the options for profitable crops are limited, which compels farmers to select crops based on considerations other than per feddan net returns and profitability.

## 4. Discussion

### 4.1. Discrepancy and Similarity between Governorates

The discrepancy between the water profitability values calculated from the face-to-face forms and the bulletin is due to several reasons. Firstly, the bulletin data were published for the year 2019, which is based on 2018 estimates and numbers, while the forms were collected in 2020. This could affect water profitability through a multitude of ways, such as different market prices and the availability of production inputs and their prices. Moreover, the bulletin data were based on averages of market prices, yields, and experts' estimations of the production inputs and thus the return per feddan. On the contrary, the value of the estimates in this study are calculated from primary collected data from farmers, thus considering the different challenges farmers may face in acquiring certain production inputs such as fertilizers and pesticides.

The estimates of water profitability from the collected are forms are higher compared to the estimates by Hosni et al. in 2013 [23]. For Sharkia Governorate their estimates of water profitability for rice were 0.32 EGP/m$^3$ compared to our estimates of 0.56 EGP/m$^3$. Sugar beet water profitability in their study was estimated at 1.68 EGP/m$^3$ compared to our estimates of 2.98 EGP/m$^3$ for the same governorate. It is also important to point out

that the estimates by Hosni et al. (ibid.) were calculated using the bulletin data and not from primary collected data from farmers.

### 4.2. Rationale for Crop Selection

Despite maize and rice having relatively lower water profitability, these crops have significant value for farmers. Maize is a major source of animal nutrition, and it provides important supplementary nutrition for livestock. The absence of these nutritious elements leads to a decrease in the production of meat, which affects Egyptian food security. Rice plays an important role in soil management; farmers plant rice to wash their soils and improve their fertility. Rice is also considered one of the most important staple foods and a source of foreign currency when exported. The shortage of this crop affects the volume of agricultural exports. Rice has another significant value as it contributes to the protection of the northern areas of the Nile Delta from seawater intrusion.

Even though cotton has also relatively lower water profitability, it is a significant Egyptian crop due to several reasons; it is an important strategic crop for the textile industry, as well as for exports [26]. Egyptian cotton is world-famous for its quality and has great export potential [27]. Many farmers responded that they continued planting cotton as they inherited the practice from their fathers and grandfathers. They also mentioned that for them there is no convincing alternative.

Wheat is an essential crop for Egyptian food security even if it has lower water profitability compared to other high-value crops [28]. Egypt is considered one of the biggest wheat importers globally; this is due to the high consumption of bread in the Egyptian diet [1,29]. Thus, many farmers in Egypt tend to grow wheat for self or home consumption. Farmers listed several other reasons for growing wheat, for example, the low amount of labor, easiness of growing the crop, having the accumulated experience and knowledge to grow it, and its usefulness as feed livestock.

Regarding the most water-profitable crops as seen in Figure 4, dry peas were the most profitable followed by watermelon pulp. Farmers justified the plantation of dry peas in Kafr El-Sheikh as it has very high net returns per feddan and relatively low costs compared to the profit. Farmers selected watermelon pulp cultivation due to the high return it generates, easiness of cultivation, and the fact that it has a short cycle so does not stay in the ground for a long period. Moreover, farmers chose broad beans because they reduce soil stress and increase its fertility, and its straw is used as fodder for livestock. These represented the secondary values that were often underestimated. Sugar beet was selected by farmers as it has high returns per feddan, it thrives in the soil in Kafr El-Sheikh, and it has a relatively stable selling price when sold to sugar factories. Finally, sugar beet can withstand salinity, which reduces the risk of growing it.

Net return and profitability are not the only factors that impact farmers' crop selection. The smaller the farm size, the fewer the options for profitable crops. However, farmers therefore tend to grow livestock on those lands and grow crops that can be used as fodder such as clover, maize, and crops that have a side product that can be used as fodder such as wheat, broad beans, and sugar beet. The net revenue and profitability of these products are relatively low, but their contribution in the value chain for farmers is high and satisfies the need for fodder for livestock, which would be expensive if purchased from the market.

### 4.3. Recommendations

The above analysis revealed the need for a new paradigm shift in the Egyptian water and food sectors in an effort to address these challenges and mitigate the risks. This paradigm has three main directions in which Egypt's water sector and food sector can transform to be able to accommodate and deal with its challenges and meet future needs, including the socio-economic development ambitions. The first dimension is the digital transformation of the agricultural sector. The second dimension is the investment in the agricultural sector and focus on its development. The third dimension is to adopt more bottom-up planning and implementation to improve equity in water access and use with

respect to agricultural water investments, which are part of a bigger picture of system management, as they are efficient from economic, social, and environmental perspectives. This entails the concept of nexus governance, requiring policy actors to engage across policy domains and the public and private spheres, and by extension, strengthening human capital and institutions for policy coherence and participatory mechanisms.

The role of education and extensions services is clear in improving land and water productivity [30]. Investment in strong extension services and awareness campaigns for farmers can significantly increase water profitability and contribute to increased levels of food security. One dimension of this could be achieved through the use of digital innovations and information systems [31]. These tools can provide farmers with accurate information and viable interventions at the right time.

The prices and availability of agricultural inputs affect the net returns per feddan and water profitability as has been found in the three governorates. Increasing the allocated quantities of seeds, fertilizers, and pesticides at the agricultural associations in each governorate and increasing the subsidies allocated to these items would positively impact water profitability for farmers, in particular the smallholders. In addition, providing machinery for farming and harvesting different crops at subsidized rates or through establishing farmers' associations could positively impact the profitability and respond to the lack of manpower and its high cost.

The next step required to better understand the agricultural system in the Nile Delta is to assess the water profitability of cropping sequences, not just single crops. Assessing common cropping rotations in the three governorates will paint a clearer picture of the small farmers in the region. Common rotations are the plantation of rice and sugar beet followed by cotton and then wheat, or starting with cotton and then wheat followed by rice and ending with wheat again [32]. Moreover, analyzing water profitability over a year, thus including every season, would take into account the same temporal scale for analyzing net return for farmers.

Assessing the water- and soil-quality effects on water profitability is essential, as it would open the door for understanding the links between the soil characteristics, land productivity, yield, and production inputs and costs. In addition, some crops are selected by farmers to improve soil fertility and to protect the land from deterioration. These links and benefits should be considered when analyzing water profitability.

Conservation agriculture is key to addressing the challenges related to food insecurity and climate change. Transformation of agricultural systems by adopting climate-smart agriculture practices can increase resilience while increasing productivity.

Finally, having a baseline of water profitability for different crops before COVID-19 could be the first step to evaluate potential new crops that have higher water profitability and can contribute directly or indirectly to improving food security in Egypt. Hence, to improve food security in Egypt, more information on crops' water profitability and their values in comparison to the world (similar countries) and region practices are essential to inform policymakers in deciding strategies regarding cropping patterns. This would create a backdrop based on which future patterns can be assessed and evaluated taking into account the pressures of climate change and economic development ambitions.

## 5. Conclusions

This study on water profitability analysis was conducted for the major crops in three governorates in Egypt. The analysis was conducted for the Sharkia, El-Beheira, and Kafr El-Sheikh governorates, situated in the Egyptian Nile Delta. The study shed light on the water profitability of different crops in the study areas based on field primary data collected from farmers in each of those governorates and verified these data through consultation workshops. This study approach has not been implemented in the Egyptian Delta before and thus reveals the actual water profitability of different crops produced by smallholder farmers. This study provides insights into the different difficulties farmers face that affect their land and water profitability and shows how these problems could be addressed to

improve food and climate security. The analysis showed the differences in water profitability among the three governorates even for the same crop and the contributing factors that affect it. Furthermore, the limiting factors for improving water profitability were identified, such as limited extension services, deteriorating water and soil quality, and inaccessibility of production inputs. Such an assessment can set the baseline for the water profitability of different crops and allow more climate-resilient cropping patterns to be planned accordingly as well as act as a guide for future policies. Monitoring the change of water profitability over time can deepen our understating of the factors that impact it along the production chain and highlight opportunities to improve it. Consequently, analyzing the water profitability of crops downstream the supply chain can paint a clearer picture of their contribution to GDP and national growth. Taking the analysis one step further and analyzing the number of family members benefiting from the generated profits could provide fresh insights into water profitability social distribution and the number of beneficiaries. Finally, we provided policy actions and recommendations for improving water profitability for farmers and future pathways for a deeper understanding of the water profitability of the farmers in the Nile Delta and how this knowledge could improve Egyptian food and climate security.

**Author Contributions:** The field research conceptualization, methodological design, and data collection was overseen and conducted by A.B., A.E., S.A.E.-H. and A.I. The data collected were analyzed by A.I. The initial figures and tables were generated by A.I. and A.B. and subsequently finalized by A.B. The paper was drafted by A.B. under the supervision of A.E., S.A.E.-H. and A.I., with final drafts generated by A.B. and finalized and edited by A.B. and A.E. Funding acquisition was carried out by A.E. All authors have read and agreed to the published version of the manuscript.

**Funding:** This project is part of the CGIAR Research Program on Water, Land and Ecosystems (WEGP) and supported by CGIAR fund donors.

**Institutional Review Board Statement:** Not applicable.

**Informed Consent Statement:** Not applicable.

**Data Availability Statement:** The data supporting the conclusions of this study are available upon reasonable request from the authors, with the exception of data that identify the personal information of the research participants.

**Acknowledgments:** This project is part of the CGIAR Research Program on Water, Land and Ecosystems (WEGP) and supported by CGIAR fund donors. The research was conducted by a team of scientists based at the International Water Management Institute–IWMI and the Soil, Water and Environmental Research Institute–SWERI, Egypt.

**Conflicts of Interest:** The authors declare no conflict of interest.

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
