# Peer review of "Water Profitability Analysis to Improve Food Security and Climate Resilience: A Case Study in the Egyptian Nile Delta"

_climate, doi:10.3390/cli10020017_

Round 1
Reviewer 1 Report
- The personal data of the authors on the first page is incomplete.
- There should be no separating space between paragraphs.
- Table 1 lacks a source and year (in addition, this source must be included in the bibliographic references).
- Table 2 lacks a source and year (in addition, this source must be included in the bibliographic references).
- Table 3 lacks a source and year (in addition, this source must be included in the bibliographic references).
- Table 4 lacks a source and year (in addition, this source must be included in the bibliographic references).
- Figure 1 lacks a source and year (in addition, this source must be included in the bibliographic references).
- Figure 2 lacks a source and year (in addition, this source must be included in the bibliographic references).
- Figure 3 lacks a source and year (in addition, this source must be included in the bibliographic references).
- Figure 4 lacks a source and year (in addition, this source must be included in the bibliographic references).
- It is necessary to deepen in the "recommendations".
- It is necessary to deepen in the "conclusions".
- I suggest citing these articles:
a) El "role-playing" en los estudios de política internacional y resolución de conflictos. REJIE: Revista Jurídica de Investigación e Innovación Educativa: https://dialnet.unirioja.es/servlet/articulo?codigo=4586001
b) https://www.worldbank.org/en/news/opinion/2021/09/24/mena-has-a-food-security-problem-but-there-are-ways-to-address-it.
Author Response
Dear Reviewer,
Thanks a lot for your valuable comments.
The following has been addressed in the new version of the manuscript:
- Personal data has been completed by adding location of each organization. Please let us know what else is missing if there is.
- All separating spaces between paragraphs have been removed.
- All tables and figures have been amended to include year and source.
- Note: Table 1,2 and 3 are from primary field data collected by the authors.
- The world bank reference has been valuable and cited in the manuscript.
- Recommendations and conclusions sections were deepened to reflect the work done.
Reviewer 2 Report
In the article „Water Profitability Analysis to improve Food Security: Case Study Egyptian Nile Delta” water profitability was conducted for three governorates in the Nile Delta in Egypt, to establish a baseline and assess the net return per unit of water of the main crops in each of these governorates. The analysis was based on extensive in-person surveys in each governorate in addition to workshop discussions with farmers. The study has highlighted the impact of lack of extension services which limits farmers’ ability to increase their land and water productivity. The article is in line with current research trends.
The article is properly structured. It contains all the parts that should be in a scientific article. The authors have conducted a review of international literature on the subject, indicated the area of research and the method of obtaining data, presented the obtained results and conducted a discussion of them, including conclusions from the analysis at the end. In terms of the structure and layout of the article I have no comments.
However, I have a series of comments that relate to the content presented in the article. I believe that certain elements should be added and some should be expanded upon.
My comments:
- In Section 1.1, please include a map showing the location of the study area. This would be a great convenience for readers, especially from other countries.
- Chapter 2.2 : it is not clear from the content presented whether the research sample is representative. The authors write that they conducted about 100 interviews with farmers. Please add how many farmers are operating in the study area, what percentage of them were included in the analysis, how farmers were selected for interview, etc.
- Please elaborate on section 2.3. This is an important scientific part of the paper. Please justify the next steps and indicate why it was taken this way and not the other way.
- Lines: 173-175 - this entry looks strange. Please post this somewhere in your research methodology.
- Chapter 3 presents results for three governorates. Please justify why different crops were analyzed at different sites.
- Lines 190, 212, 236 : the article states: Error! Reference source not found. It seems that these are references to later drawings - please correct it
- Some of the items in Chapter 3 fit more with the results of the research than with the discussion of them (e.g., lines 270-285, 287-297). Please move the elements representing the results obtained to the previous chapter, and leave the discussion alone here. What I find missing in this chapter is a reference to the results of other studies in the field and a comparison between them and the results obtained by the authors.
- In Chapter 4, please also refer to the differences between the statistical data and the data obtained by the authors from the conducted research. This is a very important element that should be developed in the discussion of the results. Why do the Authors believe that the results they obtained are better than those from the state statistics? The differences in some cases are quite large and leave doubts about the correctness of the obtained results and their representativeness.
- Please elaborate on chapter 5. it is currently written as if the authors had an imposition to make it as short as possible! Interesting, quite extensive research has been carried out, please elaborate on these elements, indicate the importance of this research, its actuality, innovativeness, describe the possibilities of use in more detail.
- Technical comment: please add under figures and tables their sources. This is a standard practice in a scientific article.
The article needs improvement and completion. It will be suitable for publication after appropriate changes have been made.
Author Response
Dear Reviewer,
Thanks a lot for your valuable comments and inputs, they are extremely appreciated.
The following has been addressed in the new version of the manuscript:
- In Section 1, a map was added showing the location of the study area.
- The numbers of total farmers in the governorate is not available, the latest estimates from the CAPMAS are for 2014. The sample was randomly selected for small holder farmers that have lands ranging from less than one feddan to five feddans. Thus, the selected crops were those identified in the sample after the farmers were selected.
- Section 2.3 was elaborated on, justifying why this approach was selected and its value.
- Lines 173-175 (regarding the units) were moved to the research methodology.
- It was added why different crops were analyzed at different sites, since the farmer sampling was random, the crops analyzed were based on the responses of the farmers.
- The error references have been amended.
- Elements mentioned have been integrated in the results, and more focused discussion on the differences between governorates, the differences between the collected data and the bulletin data and the other article published in Egypt.
- It is explained why there are differences, mainly the year of analysis and also the method of data collection. Our calculations are based on face-to-face collected forms compared to the bulletin which uses statistical averages and experts’ estimates not farmers’ real-life experiences.
- Chapter 5 has been elaborated upon to include the importance of the research and its novelty and how it differs from other studies and its potential uses and benefits.
- All figures and tables have their sources updated and provided.
Reviewer 3 Report
Comments by line:
34: suggestion: add the missing verb to the sentence: “Egypt is the largest wheat importer and…”
59: the most important adaptation strategy is the introduction of climate smart agriculture based on Conservation Agriculture and for rice the application of technical irrigation such as subsurface irrigation to create aerobic soil conditions according to the SRI concept; see also the FAO Project on Conservation Agriculture from 2007-2008 by the RRTC in Sharkia and Kafr-El-Sheikh, Nile Delta. Pls. mention in the introduction since this is also highly effective in improving water efficiency as well as avoiding salinization problems in irrigated arid areas.
190: delete the error notice and correct the sentence with the proper reference.
212: delete the error notice and correct the sentence with the proper reference.
236: delete the error notice and correct the sentence with the proper reference.
257, 260, 261, 263, Fig.4, 273: spelling of Kafr-El-Sheikh – pls. correct and make it uniform throughout the text.
326: probably the word “high” is missing in the “…very net returns…”
329: “broad” should be minuscular
345-397: the most important recommendation is missing: promote Conservation Agriculture as Climate Smart Agriculture for addressing many of the listed problems, including water productivity. Pls. add.
Author Response
Thanks a lot for your valuable comments.
The following has been addressed in the new version of the manuscript:
- Line 34 has been amended and missing verb has been added.
- Climate smart agriculture and conservation agriculture are both linked to improving water use efficiency and both points were added in the introduction and recommendation. However, we couldn’t find the report of the FAO project rom 2007-2008 by the RRTC in Sharkia and Kafr El-Sheikh, Nile Delta. It would be valuable if you could share it with us.
- 257, 260, 261, 263, Fig.4, 273: spelling of Kafr El-Sheikh. This has been amended and made consistent throughout the manuscript.
- Error notices have been removed. (They were cross-referencing and the tables and figures)
- 326: probably the word “high” is missing in the “…very net returns…” Has been amended.
- Promoting Conservation Agriculture as Climate Smart Agriculture for addressing many of the listed problems, including water productivity has been added as an important recommendation.
Round 2
Reviewer 2 Report
The authors have addressed all my comments. Their explanations satisfy me. Corrections and additions made by the Authors significantly increased the value of the article.
I recommend the article for publication.
Author Response
Thanks for your kind feedback, your comments were of great value.